# Effects of Aquatic Plant Coverage on Diversity and Resource Use Efficiency of Phytoplankton in Urban Wetlands: A Case Study in Jinan, China

**DOI:** 10.3390/biology13010044

**Published:** 2024-01-14

**Authors:** Hongjingzheng Jiang, Aoran Lu, Jiaxin Li, Mengdi Ma, Ge Meng, Qi Chen, Gang Liu, Xuwang Yin

**Affiliations:** Liaoning Provincial Key Laboratory for Hydrobiology, College of Fisheries and Life Science, Dalian Ocean University, Dalian 116023, China; jhjzzz1999@gmail.com (H.J.); luaoran1018@163.com (A.L.); shay_li@126.com (J.L.); ma1838123814@outlook.com (M.M.); mengge199805@163.com (G.M.); chenqi3663@163.com (Q.C.)

**Keywords:** wetland degradation, aquatic plants, resource use efficiency, phytoplankton diversity, ecosystem functions

## Abstract

**Simple Summary:**

An urban wetland, a critical component of an ecosystem, provides diverse habitats and has important functions such as water purification and nutrient cycling. However, under urbanization, wetland ecosystems face serious challenges from human activities like dredging, which often removes aquatic plants and destroys their functions. Therefore, studying the contributions of aquatic plants is key to wetland conservation. The present study was conducted in 10 urban wetlands in Jinan, China, to investigate the effects of aquatic plant coverage on wetland water quality, phytoplankton diversity, and resource use efficiency. The study area was categorized into three different aquatic plant coverage groups: low coverage (0–25%), medium coverage (26–35%), and high coverage (36–66%). The relationships among water quality parameters, phytoplankton diversity, and resource use efficiency were analyzed. Data show that the increase in aquatic plant coverage could directly absorb excess nutrients (e.g., nitrogen and phosphorus) and help to reduce sediment resuspension, thus significantly reducing the nutrient and suspended solid content of the water body and improving the water quality of the wetland. Furthermore, the increase in coverage was also associated with the increase in phytoplankton diversity, including species richness and functional diversity. The present study has shown that the composition of phytoplankton functional groups is positively affected by the degree of aquatic plant coverage. Phytoplankton groups adapted to still-water conditions and low light intensity were predominant in sites with higher aquatic plant coverage. Meanwhile, lower nutrients prevented dominant species from outcompeting others, allowing increased diversity. This increased phytoplankton diversity was associated with increased resource utilization efficiency (RUE), which is the ratio of phytoplankton biomass to available nutrients such as nitrogen and phosphorus. More diversity allows for better ecological niche allocation and complementarity in the utilization of limited resources. Adequate aquatic plant coverage plays a critical role in maintaining biodiversity, water quality, and ecosystem function in urban wetlands. Conservation of aquatic plants should be a priority in management plans. The results of this study provide a scientific basis for incorporating aquatic plants into sustainable urban wetland conservation strategies.

**Abstract:**

With the acceleration of urbanization, biodiversity and ecosystem functions of urban wetlands are facing serious challenges. The loss of aquatic plants in urban wetlands is becoming more frequent and intense due to human activities; nevertheless, the effects of aquatic plants on wetland ecosystems have received less attention. Therefore, we conducted field investigations across 10 urban wetlands in Jinan, Shandong Province, as a case in North China to examine the relationships between aquatic plant coverage and phytoplankton diversity, as well as resource use efficiency (RUE) in urban wetlands. Multivariate regression and partial least squares structural equation modeling (PLS-SEM) were used to analyze the water quality, phytoplankton diversity, and RUE. The results demonstrate that the increase in aquatic plant coverage significantly reduced the concentration of total nitrogen and suspended solids’ concentrations and significantly increased the phytoplankton diversity (e.g., species richness and functional diversity). The aquatic plant coverage significantly affected the composition of phytoplankton functional groups; for example, functional groups that had adapted to still-water and low-light conditions became dominant. Furthermore, the increase in phytoplankton diversity improved phytoplankton RUE, highlighting the importance of aquatic plants in maintaining wetland ecosystem functions. This study may provide a scientific basis for the management strategy of aquatic plants in urban wetlands, emphasizing the key role of appropriate aquatic plant cover in maintaining the ecological stability and ecosystem service functions of wetlands.

## 1. Introduction

A natural wetland is an ecosystem with unique hydrological, soil, vegetation, and biological characteristics distributed between terrestrial and aquatic environments, thereby providing an irreplaceable ecological function in protecting water resources, regulating climate, providing diverse habitats, and preserving biodiversity [1]. The global decline in wetland areas and the severe fragmentation of habitats have gradually lost ecosystem functions and reduced biodiversity [2]. In the past century, approximately 70% of wetland areas have decreased because of increased human activities and climate change [3]. Urban wetlands—as an important component of urban ecosystems—are more vulnerable to human-induced disturbances [4]. Phytoplankton are fundamental to material and energy flow in aquatic ecosystems and are essential for maintaining the stability and integrity of wetland ecosystems [5]. By altering the water environment, aquatic plants can influence the dynamics of phytoplankton growth and community structure, thereby affecting resource acquisition and utilization [6,7]. Therefore, understanding how aquatic plant coverage affects the diversity and function (e.g., resource use efficiency (RUE) of phytoplankton) in urban wetlands is crucial for optimizing wetland planting configuration and preserving wetland biodiversity and ecosystem services.

Aquatic plants can purify water by absorbing excess nutrients from the water body. In lake water, the roots, stems, and leaves of submerged plants absorb substantial quantities of nitrogen and phosphorus compounds, effectively controlling internal loading and reducing nitrogen and phosphorus levels, thereby reducing the eutrophication level of the water body [8,9]. Emergent plants, such as *Phragmites australis* and *Cyperus alternifolius*, have an enhanced capacity to absorb water pollutants as they possess more supporting tissues to store nutrients over longer durations [10]. These plants are vital in controlling phytoplankton [11]. Aquatic plants affect phytoplankton dynamics by competing for nutrients and light [12,13]. As nitrogen and phosphorus are crucial nutrients necessary for algal photosynthesis, metabolism, and reproduction, they are the most limiting resources for phytoplankton growth in freshwater ecosystems [14]. Aquatic ecosystems with high aquatic plant coverage readily adsorb nitrogen and phosphorus, restricting the supply of dissolved inorganic nutrients that restrain phytoplankton growth [15]. Furthermore, phytoplankton encounter an obstacle when competing for light with expansive floating plants: the substantial leaf area of the latter hinders light penetration and suppresses the photosynthesis and growth of phytoplankton [13,16].

It is critical to comprehend the dynamics of ecological systems and their response to environmental changes. A decrease in species and functional groups (FGs) within an ecosystem decreases the efficiency of capturing essential biological resources (i.e., nutrients, water, light, and prey) and converting them into biomass [17]. RUE—an important indicator for evaluating ecosystem functions—is improved in natural freshwater ecosystems because of the diversity of phytoplankton communities [18]. RUE refers to the efficiency of unit resources transformed into unit biomass, reflecting the ecosystem’s capacity to utilize and convert limiting resources [19]. Ecologists define phytoplankton RUE as the ratio of biomass (dry weight, fresh weight, or chlorophyll concentration) to limited resources, such as total nitrogen (TN) or total phosphorus (TP) [19,20]. To a certain extent, RUE eliminates the interference of differences in nutrient concentration between sites in research findings. Therefore, RUE has been widely used to explore the relationship between phytoplankton diversity and ecosystem productivity in aquatic ecosystems [21]. Studies have demonstrated that ecosystems with high species diversity and complementary ecological niches utilize resources more thoroughly and efficiently than species-poor communities [22]. Functional diversity defines the range of functional traits determining how organisms obtain resources from the environment and is crucial in deciding resource utilization [23]. High functional diversity in communities may correlate with greater resource niche partitioning, thereby potentially improving RUE within the community [24]. This finding contributes to understanding the relationship between biodiversity and ecosystem functioning.

Unlike natural aquatic ecosystems, urban wetlands are subject to more severe human disturbance and environmental pressures due to urban construction and pollution, which has negatively impacted vital ecosystem services provided by urban wetlands, such as water purification and maintenance of biodiversity. In addition, urban wetlands also shoulder social functions, including recreational and cultural services. As a result, urban wetlands’ ecological functions and conservation needs may differ significantly from those of natural aquatic systems. Previous studies have focused mainly on the influence of phytoplankton diversity on RUE in river and lake ecosystems [25]. However, research is scarce in investigating how aquatic plants in urban wetlands influence RUE by affecting the water environment and phytoplankton diversity. In urban wetlands, human activities, such as navigational dredging, often lead to unscientific removal of aquatic plants [26]. The loss of aquatic plants has severely disrupted important ecosystem services provided by wetlands, including water purification, maintenance of biodiversity, and support of ecosystem functions [27,28]. Therefore, appropriate maintenance of aquatic plant coverage in urban wetlands is critical for wetland conservation. Given the rapid urbanization in Jinan City, conducting research on urban wetland ecology and conservation has become increasingly important. The present study examined relationships among aquatic plant coverage, phytoplankton diversity indices including species richness and functional richness (FRic), and RUE using the data collected from 78 monitoring sites in 10 urban wetlands in Jinan, China. We hypothesized that (1) increased aquatic plant coverage would improve water quality by absorbing excess nutrients in the water; (2) an improvement in water quality could affect phytoplankton community structure and increase biodiversity; (3) increased phytoplankton diversity could significantly enhance phytoplankton RUE.

## 2. Methods and Materials

### 2.1. Study Area

Jinan City is located in the central region of Shandong Province, between north latitudes 36°02′–37°54′ and east longitudes 116°21′–117°93′. Mount Tai is situated to the south, whereas the city extends north across the Yellow River. The terrain slopes down gradually from south to north and showcases landforms of low mountains and hills, inclined plains, and Yellow River alluvial plains. The area features a warm temperate continental monsoon climate, characterized by four distinct seasons with an annual average temperature of 14–15 °C and a mean annual precipitation of 1039.3 mm [29]. The spring-renowned city of Jinan has devoted itself to developing water conservation initiatives [30]. The management system and ecological improvements have expanded wetland restoration. There are 391.07 hectares of wetland, including 5 national and 10 provincial wetland parks.

In September 2020 and September 2021, sampling surveys were conducted at ten wetland parks in Jinan, including five national parks (Jixi Wetland, Baiyun Lake Wetland, Rose Lake Wetland, Xueye Lake Wetland, and Dawen River Wetland) and five provincial parks (Tumahe Wetland, Chengbo Lake Wetland, Yanziwan Wetland, Dashahu Wetland, and Huashan Lake Wetland) [31]. The study also included Daming Lake Park, a location characterized by a substantial wetland ecosystem and a close relationship with human activities (Figure 1). Based on considerations of wetland area and geographical location, six sampling sites were selected in each park, with an additional three sites chosen in Daming Lake. The latitude and longitude of each sampling site were accurately recorded using the Magellan Explorist 200 GPS device.

### 2.2. Phytoplankton Collection and Functional Group Analysis

Sampling was conducted at a total of 78 stations established across 10 wetlands situated in Jinan City, Shandong Province, China. These wetlands have an average depth of 1.6 m. At each station, 1 L of water was collected at a depth of 0.5 m of water using a water sampler for a qualitative analysis. After collection, the samples were fixed with Lugol’s solution in a ratio of 100:1.5. For a quantitative analysis, triplicate samples were collected from each sampling site using 5 L glass bottles, with 1 L per replicate, and fixed with Lugol’s solution immediately after collection. Samples were transported to the laboratory in the dark and were allowed to settle for 48 h. In the laboratory, the supernatant of parallel samples was removed and condensed to 100 mL and then transferred to 100 mL plastic bottles. To quantify phytoplankton, plastic bottles were shaken vigorously 100 times horizontally. Then, 0.1 mL aliquots were transferred using a liquid transfer gun into 0.1 L phytoplankton counting frames with 10 columns and 10 rows of 2 mm × 2 mm quadrants (20 mm × 20 mm in total area) for taxonomic identification and count under a light microscope at 400× magnification. Phytoplankton species were identified following previous studies [32,33,34,35]. Cell number and biomass (wet weight) were measured using the visual field method and volumetric measurements, respectively [36]. Functional groups’ classification aims to group phytoplankton species with similar sensitivities to a given habitat type. This grouping is performed by comprehensively considering phytoplankton’s various morphological, physiological, and ecological factors [37]. The functional group classification method has been continuously revised and supplemented in its practical application, developing a more sophisticated and refined classification system [38]. The present study used the FG classification method to investigate variations in FG composition among wetland ecosystems under different aquatic plant coverages. Phytoplankton were categorized based on genus affiliation among the 39 FGs of the classification system [37,38].

### 2.3. Measurement of Water Environmental Factors and Aquatic Plant Coverage

Dissolved oxygen (DO) concentration and pH were measured on-site using the YSI ProPlus (YSI Inc, Yellow Springs, OH, USA) multi-parameter water quality meter. Water samples (2 L) were collected from each site at a depth of 0.5 m using a water sampler and then stored in an insulated box at 4 °C for a laboratory analysis. Following alkaline potassium persulfate digestion, total nitrogen (TN) and total phosphorus (TP) concentrations were quantified using UV spectrophotometry. Ammonium nitrogen (NH_4_^+^–N), nitrate nitrogen (NO_3_–N), phosphate (PO_4_^3−^–P), suspended solids (SSs), and chlorophyll a (Chl–a) were measured following the standard procedures specified by China’s EPA [39]. Photographic techniques and visual observation assessed aquatic plant coverage in wetland ecosystems. To capture images of vegetation coverage, we took two to three repeated shots at each site using a high-resolution camera. We analyzed each image to determine plant coverage, and the average value was then recorded for the specific area. A stratified sampling approach was used to investigate diverse aquatic plant communities at each site [40]. Quadruplicate samples of floating, floating-leaved, and submerged plants were collected through a hand net measuring 0.5 m × 0.5 m. The total biomass was calculated by determining the fresh weight of all plants within every quadrant and dividing by the unit area. At every site, four 2 m × 2 m quadrants were chosen randomly to gauge the plant’s above-ground biomass (fresh weight). The unit area’s biomass was then calculated for the emergent plant communities. Based on the range of aquatic plant coverage, the sampling sites were classified into three groups, the low coverage group (LCG), the medium coverage group (MCG), and the high coverage group (HCG) (Table 1), with the aquatic plant coverage in each group ranging from 0 to 25% (*n* = 32), 26 to 35% (*n* = 24), and 36 to 66% (*n* = 22), respectively.

### 2.4. Calculation of Phytoplankton Diversity and Dominance Degrees

The dominance degree provides insight into the roles of each species within the community, summarizing the trait values and ranges that influence ecosystem functions. Functional diversity is gaining prominence as a description of phytoplankton diversity [41]. This study created a trait matrix using ten functional traits associated with phytoplankton’s critical morphological, physiological, and behavioral features to determine functional richness (FRic) [42] (Table 2). Calculation of phytoplankton dominance was conducted as described in [43].

### 2.5. Phytoplankton Resource Use Efficiency

Phytoplankton RUE is the ratio of phytoplankton biomass to available resources [20,21]. Nitrogen and phosphorus limit the growth of phytoplankton in most freshwater ecosystems. These primary nutrients are typically used to determine phytoplankton RUE using the following formulas:(1)$$RUE\_TN=\log_{10}(\frac{Biomass}{\rho \left(TN\right)}+1)$$
(2)$$RUE\_TP=\log_{10}(\frac{Biomass}{\rho \left(TP\right)}+1)$$

Phytoplankton biomass is measured in mg/L, and ρ(TN) and ρ(TP) represent the total nitrogen (TN) and total phosphorus (TP) concentrations (mg·L^−1^), respectively. RUE_TN and RUE_TP represent the phytoplankton utilization efficiencies for TN and TP, respectively.

### 2.6. Data Analysis

A Kruskal–Wallis test was used to compare differences in aquatic environmental variables and the coverage of various aquatic plant categories among coverage groups. A radial stacked bar plot was created in the software Origin 2023 (Origin Lab, Northampton, MA, USA) to display changes in the functional group biomass for different levels of aquatic plant coverage. Phytoplankton functional richness was calculated using the “FD” package in R-4.2.2 software [46,47]. After conducting a square root transformation of the functional richness, we performed a single-factor ANOVA in Origin 2023 to analyze statistical differences in species richness and functional diversity between the coverage groups. Doughnut plots, illustrating the abundance and biomass of phytoplankton, were generated using Origin 2023. Based on our research objectives and a comprehensive literature review, we adopted the stepwise forward selection strategy and selected five environmental variables for a further analysis. Compared to the conventional Structural Equation Model (SEM), PLS-SEM is generally better suited for small sample sizes. Therefore, we developed four latent variables—aquatic plant coverage, environmental factors, and phytoplankton diversity—to investigate the direct and indirect effect of plant coverage on phytoplankton RUE through PLS-SEM. A multiple regression analysis explored the relationship between aquatic plant coverage, environmental factors, phytoplankton diversity, and RUE. The observed variables for aquatic plant coverage comprised submerged, floating, floating-leaved, and emergent plant coverage. The observed environmental factors were water depth (WD), TN, Chl-a, SS, and PO_4_^3^–P. Furthermore, species richness and functional diversity were the observed variables for the diversity index. These variables were analyzed using the “plspm” and “tidyverse” packages in the R software [48,49].

## 3. Results

### 3.1. Physicochemical Characteristics in Different Aquatic Plant Coverage Groups

The ANOVA results show significant differences among coverage groups in several indicators (Table 3). The coverage of submerged and floating plants exhibited no significant differences among the groups. Emergent plants were the dominant species in all cover groups, with an average coverage ranging from 8.7% to 31.7%. The highest coverage of floating-leaved plants was observed in the MCG, with 4.6 ± 6%. The concentration of TN decreased to its minimum level in the MCG (2.595 ± 2.799), which was significantly lower than in the LCG (*p* = 0.029; Kruskal–Wallis test). Additionally, the NO_3_–N trend was similar to that of TN, reaching the lowest value in the MCG (1.615 ± 1.865), which was significantly lower than the other two groups (*p* = 0.026). The statistical differences in TN and NO_3_–N concentrations among groups imply that aquatic plant coverage may influence nutrient cycling in water bodies.

### 3.2. Phytoplankton Community Composition and Functional Groups

A total of 255 phytoplankton species belonging to eight phyla were identified across the wetlands. Among them, Chlorophyta (87 species, 34%), Bacillariophyta (74 species, 29%), and Cyanobacteria (45 species, 18%) were the dominant phyla. At different levels of aquatic plant coverage, Chlorophyta and Bacillariophyta were the dominant phytoplankton taxa. The low coverage group had a total of 178 species, with Chlorophyta accounting for 31% and Bacillariophyta for 30%. The medium coverage group had 148 species, with 31% Chlorophyta and 28% Bacillariophyta. The high coverage group had 132 species, with Chlorophyta and Bacillariophyta representing 39% and 27%, respectively. In Figure 2, box plots show the differences in species richness and FRic among the different coverage groups. The one-way analysis of variance (ANOVA) showed species richness and FRic were significantly higher in MCG and HCG than in LCG, while there was no significant difference between MCG and HCG. As shown in Table 4, cyanobacteria were overwhelmingly abundant in urban wetlands. Cyanobacteria, particularly *Phormidium tenue*, dominated all three aquatic coverage groups. In urban wetlands, phytoplankton can be classified into 28 functional groups, named A, B, C, D, F, G, H1, J, K, LM, LO, M, MP, N, NA, P, S1, S2, SN, T, TB, TC, W1, W2, X1, X2, X3, and Y, each corresponding to unique genera and habitat characteristics as shown in Table 5. We defined the phytoplankton functional groups with a relative biomass of more than 5% as the dominant functional groups in this period. In the LCG, the dominant functional groups were C, D, J, MP, S1, and TC, with S1 and TC accounting for 19.18% and 32.66%, respectively. The predominant functional groups in the MCG were D, S1, and TC, accounting for 5.37%, 36.74%, and 42.65%, respectively (Figure 3). In contrast, the HCG was dominated by S1, TC, and Y, accounting for 12.31%, 46.71%, and 19.09%, respectively. With the gradual increase in coverage, the proportion of functional group TC increased significantly, establishing its dominance in all three groups. Thus, this functional group emerged as the dominant force in the urban wetland.

### 3.3. Differences in Species Abundance and Biomass of Phytoplankton by Category among Different Aquatic Plant Coverage Groups

Figure 4 comprehensively presents the proportion of phytoplankton species abundance (a) and biomass (b) for different phyla across aquatic plant coverage groups. The results showed no significant change in the proportion of cyanobacterial species abundance between the groups. The MCG exhibited a higher cyanobacterial biomass proportion of 72.1%, attributable to the prolific growth of *Phormidium tenue*. Furthermore, Bacillariophyta species richness and biomass decreased accordingly when aquatic plant cover exceeded 35%. Specifically, the percentage of species abundance reduced from 30.1% to 21.2%, while the corresponding biomass proportion decreased from 28.5% to 9.3%. However, no significant difference in the abundance of Chlorophyta species was observed between the coverage groups. The decrease in the abundance and biomass of Bacillariophyta increased the percentage of Chlorophyta biomass from 30.7% to 41.2%. Also, the proportion of Cryptophyta biomass increased from 3.4% to 21.4%.

### 3.4. Results of Multiple Regression Analysis

The Table 6, Table 7 and Table 8 presents the associations between variables determined from the multiple regression analysis. It encompassed the influence of aquatic plant coverage on environmental factors (Table 6), aquatic plant coverage and environmental factors on phytoplankton diversity, and aquatic plant coverage (Table 7), environmental factors, and phytoplankton diversity on RUE (Table 8). Table 6 shows the different influences of four aquatic plant types on environmental factors. Emergent plants significantly reduced the TN concentration in the water body, while submerged and floating-leaved plants also showed negative correlations, but with non-significant effects. In particular, emergent plants significantly reduced suspended solids, and the presence of floating-leaved plants was associated with increased phosphate in the water column. Table 7 shows that enhancing aquatic plant coverage and improving the water environment can increase phytoplankton diversity. Aquatic plant coverage displayed a significant and positive correlation with both the taxonomic and functional diversity of phytoplankton. The aquatic plant assemblage accounted for 14.5% of the variation seen in functional diversity. Furthermore, the combination of environmental factors was identified as the primary factor driving biodiversity change, explaining 38.3% and 38.9% of the variance in species richness and functional diversity, respectively. The results showed that increased concentrations of Chl-a and phosphate concentrations as well as decreased SS were associated with higher species richness, whereas lower TN was associated with higher functional diversity. Excessive TN may lead to dominant species overgrowth, thus inhibiting other populations. Table 8 shows that water environmental factors and biodiversity significantly affect the RUE of phytoplankton. Environmental factors account for 43.9% and 15.3% of the variation observed in RUE_TN and RUE_TP, while biodiversity explains 23.6% and 8.5%, respectively. TN was negatively correlated with RUE_TN, whereas Chl-a, phosphate, and functional diversity positively influenced it. This suggests that managing nitrogen sources and increasing diversity may optimize nitrogen resource use.

### 3.5. The PLS-SEM Model

Figure 5 illustrates the pathways to investigate the effects of aquatic plant coverage on RUE using PLS-SEM. The total effect of aquatic plant coverage on RUE was 0.168, and the direct effect on RUE was insignificant. Instead, RUE improvement primarily resulted from enhancing water quality and phytoplankton diversity. The PLS-SEM analysis revealed that submerged plants had a total impact effect of 0.218 on RUE, mainly through direct effects.

In contrast, the total impact effects of floating-leaved plants and floating plants on RUE were 0.096 and −0.021, respectively, which were smaller and non-significant, probably because of their lower coverage in the wetland. The direct effect of aquatic plant coverage on RUE was negative (−0.192), but the positive indirect effect (0.252) was greater than the negative direct effect, resulting in an overall effect of 0.06. Different aquatic plants had different effects on RUE, with submerged aquatic plants having the most significant effect on increasing phytoplankton RUE.

## 4. Discussion

The present study showed that an increase in the total coverage of aquatic plants significantly lowered the concentrations of TN and SS in the water body, creating favorable habitats for phytoplankton. Therefore, phytoplankton species richness and functional diversity were enhanced, increasing RUE. As primary producers and water purifiers in wetland ecosystems, plants play a critical role in regulating the ecological balance and development of wetlands [50]. Aquatic plants effectively remove excess nutrient salts such as nitrogen and phosphorus from the water [51]. They also effectively purify water quality by intercepting and immobilizing suspended particles and inhibiting the resuspension of sediments through tissues such as the root system. Furthermore, plants can suppress excessive algal growth by competing with phytoplankton for living space, light, and nutrients or by secreting allelopathic compounds that inhibit algae [52,53]. Considering these factors, the increase in aquatic plant coverage significantly improved water quality, increased phytoplankton diversity, and further improved phytoplankton RUE.

The growth of aquatic plant communities has a non-linear effect on ecosystem energy. Our results indicate that aquatic plants in wetlands can significantly reduce the levels of TN and SS in water bodies while serving as an important means of removing excess nutrients and purifying overall water quality. This finding is consistent with previous research. Aquatic plants in wetlands have the potential to eliminate excess inorganic nutrients. Madsen and Cedergreen [54] reported that the roots and leaves of submerged plants could absorb high levels of nutrients. For instance, *Vallisneria natans* exhibited purification effects of 12.16% on nitrogen and 92.98% on phosphorus [55]. Similarly, Rong et al. found that *Potamogeton distinctus* could remove TP and total dissolved phosphorus from water [56]. Emergent plants are advantageous in light competition, allowing them to obtain nutrients from sediments and the water column. During the growth process, aquatic plants continuously absorb nutrients, such as nitrogen and phosphorus, from the surrounding environment and accumulate the nutrients in their organs, including roots, stems, and leaves. The stems and leaves of aquatic plants often protrude from the water, making it easy to harvest and remove the accumulated nitrogen and phosphorus from the plant tissues. For instance, aquatic plant assemblages have been used successfully in park pond restoration [57]. Floating-leaved plants possess robust root systems, which can extract excess nutrients from water bodies. *Lythrum salicaria* is known for its efficient nitrogen and phosphorus removal capabilities [58]. *Ipomoea aquatica* has exceptional wastewater nitrogen and phosphorus removal efficacy [59]. Without roots, floating aquatic plants with rapid growth rates float on the water’s surface. *Hyacinthus orientalis* and *Tagetes erecta* are floating plants that absorb diverse heavy metal pollutants in water bodies and are highly valuable in water purification [60]. The buffering layer of emergent plants reduced wave disturbance in the water, improving conditions for the sedimentation of SS and reducing the possibility of resuspension [57]. Studies have indicated that floating-leaved plants significantly reduced SS to increase water transparency [16,61]. However, our study did not find this to be the case.

The increase in aquatic plant coverage had a significant effect on the water environment and the structure of the phytoplankton community. The phytoplankton functional groups in the LCG were mostly those that preferred turbid, shallow water with frequent stirring and adaptation to low temperatures, such as the functional group MP represented by *Oscillatoria* spp., the functional group J dominated by *Scenedesmus* spp., the functional group D represented by *Synedra* spp., and the functional group C represented by *Aulacoseira* spp. and *Cyclotella* spp. With the increase in aquatic plant coverage, the light intensity was further decreased, causing the functional group Y, characterized by a relatively high surface area and adaptation to low-light conditions, to emerge as the dominant group in the HCG [62,63], with *Cryptomonas* spp. and *Gymnodinium* spp. being the primary representative genera. Nutrients, along with light, are crucial for the growth of phytoplankton. Nutrient deficiencies can affect phytoplankton growth, reproduction, and community structure. According to a previous study [64], an N:P ratio of around 16:1 was optimal for phytoplankton growth. The present study found an average N:P ratio of 60:1 in the evaluated wetlands (N = 3.28 mg/L and P = 0.05 mg/L). Figure 4 indicates that the absorption of nitrogenous nutrients by aquatic plants significantly increased when their coverage exceeded 35%. The species abundance and biomass of Bacillariophyta decreased with the decrease in the N:P ratio from 71.4 in the LCG to 45.9 in the HCG. Previous research has indicated that high N:P ratios (64:1 and 128:1) have a positive effect on the growth of Bacillariophyta [65], possibly because added nitrogen and phosphorus facilitate the uptake of silicates by Bacillariophyta to promote their growth. Additionally, increased coverage leads to a higher proportion of Cryptophyta biomass. Cryptomonas belongs to functional group Y, which thrives in low-light, still-water environments with high coverage.

Our study demonstrated that apart from emergent plants, other aquatic plants in urban wetlands did not significantly affect phytoplankton diversity. Increased emergent plant coverage significantly elevated phytoplankton species richness and functional diversity. Emergent plants absorb excess nitrogen in water and reduce SS, thus improving the living environment for phytoplankton and enhancing their diversity [66]. The PLS-SEM path analysis demonstrated a significant role of water environmental factors in influencing phytoplankton diversity, with TN concentration being a decisive factor. Regression analysis results revealed that TN had a significantly negative effect on phytoplankton functional diversity, possibly because excessive nitrogen concentrations can disrupt the nutrient ratio in wetlands, allowing for the rapid dominance of certain species, such as cyanobacteria, which are highly adaptable to the environment [67]. Consequently, non-dominant species may experience a gradual loss of nutrient and survival space, ultimately decreasing the community diversity and homogenization of the functional phytoplankton community. Decreased nitrogen concentration provides a survival environment for some non-dominant species, causing algae to adopt different growth strategies and increasing the overall functional diversity of the community. Reduced SS alters underwater light conditions, increases transparency, relieves light limitation, and increases the abundance of species well adapted to light [68]. Besides TN, phosphate is also an important variable affecting phytoplankton diversity. Results presented in Table 7 indicate that phosphorus significantly affects both phytoplankton species richness and functional diversity. Previous studies have suggested that inorganic orthophosphate is essential for algae’s energetic and biochemical processes and is the preferred phosphorus source for algae growth, particularly for cyanobacteria. In this wetland, algae quickly assimilate and utilize increased levels of inorganic orthophosphate, creating nutritional conditions for the rapid growth of cyanobacteria, which is the limiting factor for algal growth [64]. The findings suggest that reducing phosphorus limitation has a positive effect on phytoplankton growth within this wetland.

In the present study, the PLS-SEM model demonstrated a remarkable positive correlation between species diversity and RUE, consistent with other studies [20]. For instance, Ptacnik and coworkers analyzed more than 3000 phytoplankton samples from lakes in Finland, Norway, Sweden, and Denmark and found that phytoplankton with high diversity had a greater ability to utilize limited resources [20]. Similarly, Striebe and his team reported a positive correlation between phytoplankton species richness and RUE in natural aquatic ecosystems [69]. An increasing number of studies have shown that the effects of diversity on ecosystem functioning largely depend on species traits and functional roles [70,71]. Therefore, functional diversity has been proposed to improve mechanistic understanding of the relationship between biodiversity and ecosystem functioning [42]. Indeed, several studies have shown that trait-based functional diversity is superior to species diversity in predicting ecosystem functioning [72]. The multiple regression analysis in the present study also showed that functional diversity outperformed species richness in improving RUE, possibly because communities with more diverse traits have better ecological niche differentiation [70,73]. Furthermore, ecological niche complementation provides physiological advantages for more efficient use of resources, allowing for more optimal functioning of diverse ecosystems [74].

## 5. Conclusions

During the management process in urban wetlands, increasing the proportion of submerged aquatic plants is crucial while maintaining a certain level of aquatic plant coverage. Future research should focus on community structures, spatial distribution patterns of aquatic plants in wetlands, and their allelopathic effects. More comprehensive surveys are needed to characterize assemblages of aquatic plants occupying different wetland habitat types, elucidating their community characteristics and environment relationships. In addition, the massive attachment and excessive growth of benthic microalgae and cyanobacteria could inhibit the growth of aquatic plants. Certain cyanobacteria may release cyanotoxins, which can have toxic effects on plants through direct contact or transportation pathways. However, the mechanisms and extent of the ecological impact on aquatic and phytoplankton plants require further elucidation. Thus, a quantified analysis of aquatic plant distributions and effects would inform management decisions and provide insights into underlying mechanisms, which govern the role of aquatic flora in maintaining wetland ecosystem functions and may facilitate more prudent management strategies to conserve vulnerable aquatic ecosystems.

## Figures and Tables

**Figure 1 biology-13-00044-f001:**
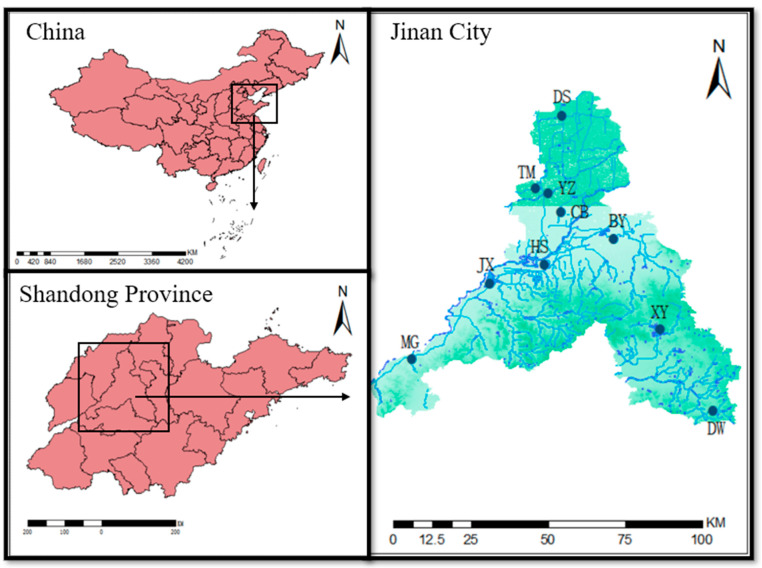
The distribution of 10 wetland parks in Jinan, Shandong Province, China. JX—Jixi Wetland, BY—Baiyun Lake Wetland, MG—Rose Lake Wetland, XY—Xueye Lake Wetland, DW—Dawen River Wetland, TM—Tumahe Wetland, CB—Chengbo Lake Wetland, YZ—Yanziwan Wetland, DS—Dashahu Wetland, and HS—Huashan Lake Wetland.

**Figure 2 biology-13-00044-f002:**
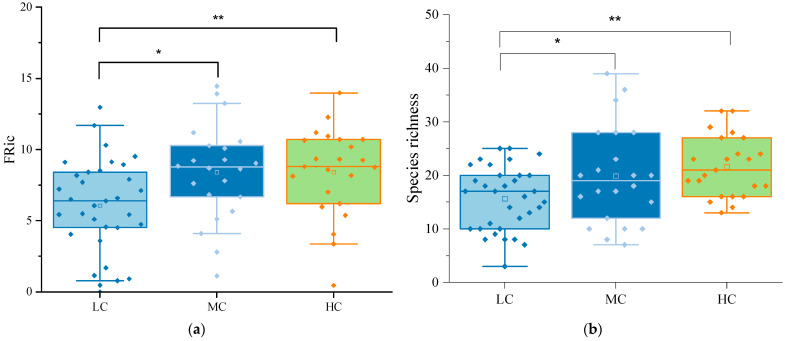
(**a**) Box plots showing functional diversity (FRic) in the low, medium, and high coverage groups. (**b**) Box plots showing taxonomic diversity (species richness, SR) in the low, medium, and high coverage groups. One-way ANOVA reveals significant differences in both indices across varying coverage levels at * *p* < 0.05 and ** *p* < 0.01.

**Figure 3 biology-13-00044-f003:**
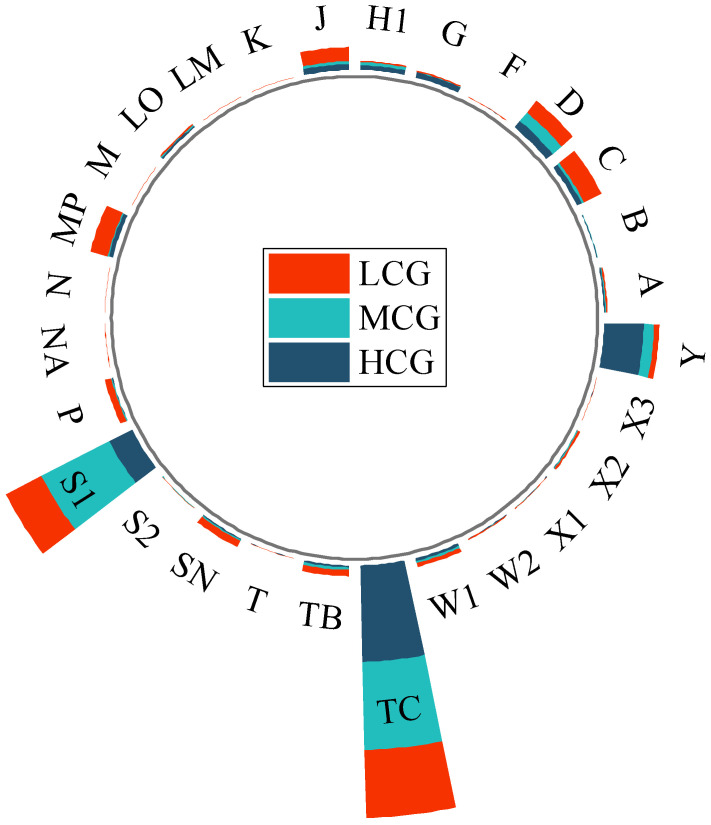
Radial stacked bar chart of phytoplankton functional group distribution under different coverage levels. The bars are stacked from the inside out in the order of low, medium, and high coverage. The functional groups in the diagram are described in Table 4.

**Figure 4 biology-13-00044-f004:**
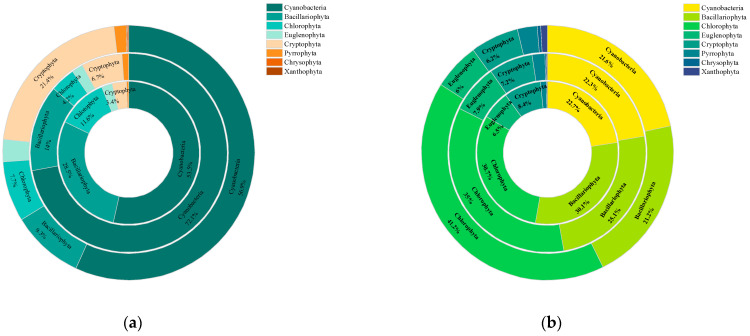
Doughnut plots of phytoplankton species abundance (**a**) and phytoplankton biomass (**b**) among different aquatic plant coverage groups, from the inner to the outer, for groups LCG, MCG, and HCG, respectively. Species category labels and values for species occupying less than 3% of the total are omitted.

**Figure 5 biology-13-00044-f005:**
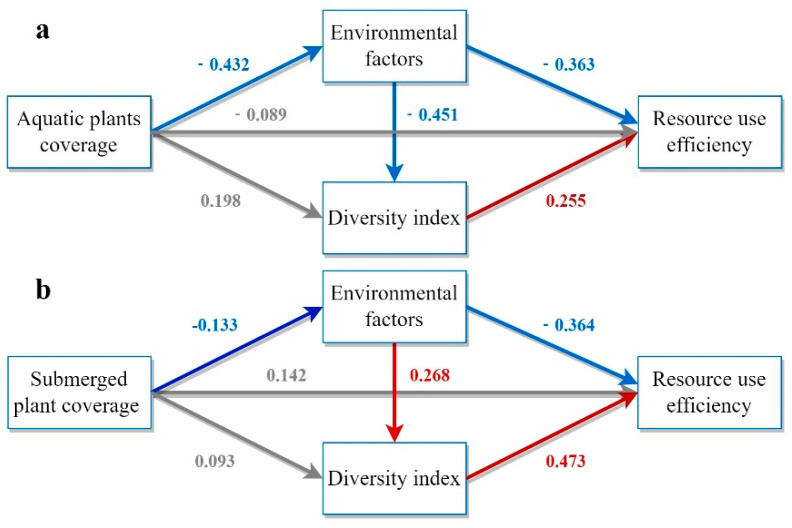
Model demonstration of the impact of different categories of aquatic plants on RUE. (**a**) Model of the influence paths of aquatic plant coverage on RUE based on PLS−SEM. (**b**) Model of the influence paths of submerged plant coverage on RUE based on PLS−SEM. (**c**) Model of the influence paths of floating−leaved plant coverage on RUE based on PLS−SEM. (**d**) Model of the influence paths of emergent plant coverage on RUE based on PLS−SEM. (**e**) Model of the influence paths of floating plant coverage on RUE based on PLS−SEM. The model includes three indirect influence paths and one direct influence path. Rectangles represent latent variables including aquatic plant coverage, environmental factors, diversity indices, and RUE. Arrows indicate hypothetical relationships between latent variables. Grey, red, and blue arrows denote non−significant, significantly positive, and significantly negative correlations, respectively. Path coefficients are displayed on the arrows.

**Table 1 biology-13-00044-t001:** Number of samples and groups of aquatic vegetation coverage. Classes: low coverage = 0–25%; medium coverage = 26–35%; high coverage = 36–66%.

Coverage (%)	Range (%)	Sample (*n*)
Low coverage group (LCG)	0–25%	32
Medium coverage group (MCG)	26–35%	24
High coverage group (HCG)	36–66%	22

**Table 2 biology-13-00044-t002:** Traits of phytoplankton according to [44,45].

Traits	Categories	Code
Morphological traits		
Biovolume (μm^3^)	<100, 100–1000, 1000–10,000, >10,000	Sma, Med, Lar, Xla
Greatest axial linear dimension (GALD)	<35 μm or >35 μm	Gal
Life form	Single-celled, colonial, filamentous	Sin, Col, Fil
Behavioral traits		
Motility	Presence/absence of flagella	Fla
Vacuolated	Yes/no	Vac
Physiological traits		
N2 fixation	Yes/no	N2f
Si requirements	Yes/no	Sil
Mixotrophy (phagotrophy)	Yes/no	Mix
Heterotrophy	Yes/no	Het
Pigment composition	Chl-b, Chl-c, phycobiliproteins	ChlB, ChlC, Phy

**Table 3 biology-13-00044-t003:** Comparison of environmental variables and aquatic plants among low, medium, and high coverage groups (mean ± SD). Bold values indicate significant differences in variables across coverage levels (Kruskal–Wallis test: *p* < 0.05).

Variable	Low Coverage Group	Medium Coverage Group	High Coverage Group	*p*	χ^2^
Total aquatic plant coverage (%)	0.127 ± 0.084	0.273 ± 0.041	0.42 ± 0.087	**<0.001**	63.018
Submerged plant coverage (%)	0.023 ± 0.044	0.046 ± 0.053	0.036 ± 0.053	0.106	4.486
Floating plant coverage (%)	0.006 ± 0.017	0.009 ± 0.038	0.029 ± 0.064	0.132	4.047
Floating-leaved plants	0.01 ± 0.027	0.046 ± 0.06	0.038 ± 0.06	**0.004**	10.941
Emergent plant coverage (%)	0.087 ± 0.074	0.171 ± 0.077	0.317 ± 0.099	**<0.001**	44.521
Water depth (WD) (m)	1.658 ± 0.668	1.518 ± 0.833	1.5 ± 0.769	0.557	1.172
pH	8.139 ± 0.452	8.268 ± 0.508	8.111 ± 0.336	0.492	1.42
Electrical conductivity (EC) (s/cm)	1127.424 ± 829.96	1149.455 ± 718.367	890.5 ± 499.538	0.519	1.31
Dissolved oxygen concentration (DO) (mg/L)	8.267 ± 2.525	8.75 ± 1.565	8.706 ± 1.355	0.768	0.527
Ammonium nitrogen (NH_4_^+^–N) (mg/L)	0.279 ± 0.397	0.153 ± 0.182	0.167 ± 0.153	0.201	3.208
Nitrate nitrogen (NO_3_–N) (mg/L)	3.018 ± 2.274	1.615 ± 1.865	1.738 ± 2.051	**0.026**	7.286
Total nitrogen (TN) (mg/L)	4.175 ± 2.818	2.595 ± 2.799	2.92 ± 3.214	**0.029**	7.109
Total phosphorus (TP) (mg/L)	0.065 ± 0.037	0.051 ± 0.023	0.051 ± 0.016	0.431	1.681
Phosphate (PO_4_^3−^–P) (mg/L)	0.029 ± 0.043	0.015 ± 0.014	0.014 ± 0.024	0.099	9.047
Suspended solids (SSs) (mg/L)	81.091 ± 119.642	31.182 ± 27.150	20.056 ± 18.574	0.178	3.454
Chlorophyll-a (Chl–a) (μg/L)	0.014 ± 0.01	0.011 ± 0.007	0.011 ± 0.007	0.474	1.491

**Table 4 biology-13-00044-t004:** Composition of dominant species in different coverage groups.

Coverage	Dominance	Dominant Species
Low coverage group (LCG)	0.153	*Phormidium tenue*
Medium coverage group (MCG)	0.099	*Phormidium tenue*
	0.025	*Merismopedia tenuissima*
High coverage group (HCG)	0.071	*Phormidium tenue*
	0.097	*Anabaena circinalis*

**Table 5 biology-13-00044-t005:** Habitat template and representative species for the functional groups of phytoplankton taxa in the studied wetland ecosystem.

Code	Habitat Template	Representative Genus/Species
A	Clear, deep-water oligotrophic lakes, usually well-mixed and phosphorus-deficient	*Cyclotella comensis*
B	Mesotrophic, small to large, shallow lakes with vertical mixing	*Cyclotella* spp.
C	Eutrophic small and medium lakes	*Cyclotella meneghiniana*
D	Shallow, eutrophic, well-aerated waters, typically turbid	*Synedra* spp., *Nitzschia* spp.
F	Clear mesotrophic lakes	*Dictyosphaerium* spp., *Kirchneriella* spp., *Oocystis* spp.
G	Small, eutrophic, still lakes	*Eudorina* spp., *Pandorina* spp.
H1	Eutrophic, both stratified and shallow lakes with low nitrogen content	*Anabaena flos-aquae*, *Anabaena circinalis*
J	Eutrophic shallow freshwaters, including low-gradient rivers	*Pediastrum* spp., *Coelastrum* spp., *Crucigenia* spp., *Scenedesmus* spp.
K	Eutrophic shallow water	*Aphanocapsa* spp., *Aphanothece* spp.
L_M_	Small–medium eutrophic–hypereutrophic, low-carbon waters	*Microcystis* spp.
L_O_	Stratified mesotrophic lakes	*Peridinium* spp., *Merismopedia* spp., *Ceratium* spp., *Ceratium* spp.
M	Eutrophication to severe eutrophication, small- and medium-sized water bodies	*Microcystis* spp.
MP	Frequently churned, turbid, shallow lakes	*Cocconeis* spp., *Dictyosphaerium* spp., *Surirella* spp. *Nitzschia* spp., *Chlorococcum* spp., *Oscillatoria* spp.
N	Summer in low-latitude or temperate lakes	*Cosmarium* spp.
NA	Poor to mesotrophic, hydrostatic, low-latitude regions	*Cosmarium* spp.
P	Eutrophic low-latitude or temperate lakes	*Melosira* spp., *Closterium* spp., *Staurastrum* spp.
S1	Turbid mixed environments	*Phormidium* spp., *Lyngbya* spp.
S2	Warm, shallow, highly alkaline waters	*Spirulina* spp.
SN	Warm mixed epilimnia	*Raphidiopsis* spp.
T	Continuously mixed epilimnia	*Tribonema* spp.
TB	Highly lotic environments, rapids	*Achnanthes* spp., *Fragilaria* spp., *Gomphonema* spp., *Melosira varians*, *Navicula* spp., *Nitzschia* spp., *Surirella* spp.
TC	Eutrophic lentic waters, or low-gradient lotic systems	*Oscillatoria* spp., *Phormidium* spp.
W1	Shallow waters with organic pollution	*Euglena* spp., *Phacus* spp., *Lepocinclis* spp.
W2	Mesotrophic pools, temporary shallow lakes	*Trachelomonas* spp., *Strombomonas* spp.
X1	Eutrophic shallow waters	*Ochromonas* spp.
X2	Moderately eutrophic to eutrophic shallow waters	*Chrysocromulina* spp.
X3	Shallow, clean mixed water bodies	*Schroederia* spp., *Chlorella* spp., *Chromulina* spp.
Y	Medium to eutrophic, low-light still-water bodies	*Cryptomonas* spp., *Teleaulax* spp., *Komma* spp., *Gymnodinium* spp., *Glenodinium* spp.

**Table 6 biology-13-00044-t006:** Results of multiple regression analysis for the effect of aquatic plant coverage on environmental factors.

	Variable	Regression Coefficient	Standard Error	*t*-Value	*p*-Value	R^2^	*p*-Value
Water depth (WD)	Submerged plants	−0.089	0.124	−0.716	0.476	0.028	0.729
	Floating-leaved plant	−0.054	0.121	−0.445	0.658		
	Emergent plant	0.05	0.12	0.417	0.678		
	Floating plant	−0.112	0.117	−0.955	0.343		
Total nitrogen (TN)	Submerged plants	−0.136	0.118	−1.15	0.254	0.111	0.073
	Floating-leaved plant	0.026	0.115	0.224	0.823		
	Emergent plant	−0.307	0.114	−2.68	0.009		
	Floating plant	0.149	0.112	1.332	0.187		
Chlorophyll-a (Chl-a)	Submerged plants	0.163	0.122	1.332	0.187	0.048	0.461
	Floating-leaved plant	−0.119	0.119	−0.994	0.323		
	Emergent plant	0.106	0.118	0.893	0.375		
	Floating plant	−0.115	0.116	−0.991	0.325		
Suspended solids (SSs)	Submerged plants	0.003	0.114	0.027	0.978	0.18	0.006
	Floating-leaved plant	−0.154	0.111	−1.388	0.17		
	Emergent plant	−0.407	0.11	−3.701	<0.001		
	Floating plant	0.009	0.107	0.08	0.937		
Phosphate (PO_4_^3−^–P)	Submerged plants	−0.089	0.121	−0.743	0.46	0.066	0.29
	Floating-leaved plant	0.263	0.118	0.222	0.03		
	Emergent plant	−0.002	0.117	−0.015	0.988		
	Floating plant	−0.029	0.114	−0.261	0.795		

**Table 7 biology-13-00044-t007:** Results of multiple regression analysis for the effect of aquatic plant coverage and environmental factors on phytoplankton diversity.

	Variable	Regression Coefficient	Standard Error	*t*-Value	*p*-Value	R^2^	*p*-Value
Species richness (SR)	Submerged plants	0.133	0.116	1.147	0.255	0.048	0.197
	Floating-leaf plant	0.024	0.113	0.214	0.831		
	Emergent plant	0.392	0.112	3.502	<0.001		
	Floating plant	−0.021	0.109	−0.191	0.849		
FRic	Submerged plants	0.141	0.116	1.218	0.227	0.145	0.022
	Floating-leaf plant	−0.011	0.113	−0.1	0.921		
	Emergent plant	0.384	0.112	3.418	0.001		
	Floating plant	−0.075	0.11	−0.683	0.497		
Species richness (SR)	Water depth (WD)	0.186	0.094	1.98	0.052	0.383	<0.001
	Total nitrogen (TN)	−0.042	0.103	−0.405	0.686		
	Chlorophyll-a (Chl-a)	0.353	0.094	3.743	<0.001		
	Suspended solids (SSs)	−0.284	0.098	−2.9	0.005		
	Phosphate (PO_4_^3−^–P)	0.366	0.1	3.673	<0.001		
FRic	Water depth (WD)	0.186	0.093	2.005	0.049	0.399	<0.001
	Total nitrogen (TN)	−0.459	0.102	−4.514	<0.001		
	Chlorophyll-a (Chl-a)	0.323	0.093	3.467	0.001		
	Suspended solids (SSs)	−0.071	0.097	−0.737	0.463		
	Phosphate (PO_4_^3−^–P)	0.324	0.098	3.292	0.002		

**Table 8 biology-13-00044-t008:** Results of multiple regression analysis for the effect of aquatic plant coverage, environmental factors, and phytoplankton diversity on resource use efficiency.

	Variable	Regression Coefficient	Standard Error	*t*-Value	*p*-Value	R^2^	*p*-Value
RUE_TN	Submerged plants	0.168	0.091	1.848	0.069	0.072	0.242
	Floating-leaf plant	0.021	0.089	0.241	0.81		
	Emergent plant	0.138	0.088	1.565	0.122		
	Floating plant	−0.046	0.086	−0.536	0.593		
RUE_TP	Submerged plants	0.047	0.029	1.636	0.106	0.06	0.339
	Floating-leaf plant	0.013	0.028	0.46	0.647		
	Emergent plant	−0.017	0.028	−0.625	0.534		
	Floating plant	0.005	0.027	0.198	0.844		
RUE_TN	Water depth (WD)	−0.069	0.068	−1.012	0.315	0.439	<0.001
	Total nitrogen (TN)	−0.461	0.074	−6.218	<0.001		
	Chlorophyll-a (Chl-a)	0.203	0.068	2.982	0.004		
	Suspended solids (SSs)	0.107	0.07	1.515	0.134		
	Phosphate (PO_4_^3−^–P)	0.163	0.072	2.278	0.026		
RUE_TP	Water depth (WD)	−0.051	0.026	−1.964	0.053	0.153	0.034
	Total nitrogen (TN)	−0.03	0.029	−1.052	0.297		
	Chlorophyll-a (Chl-a)	0.057	0.026	2.199	0.031		
	Suspended solids (SSs)	0.002	0.027	0.069	0.945		
	Phosphate (PO_4_^3−^–P)	0.04	0.028	1.462	0.148		
RUE_TN	Species richness (SR)	−0.031	0.112	−0.275	0.784	0.236	<0.001
	FRic	0.389	0.112	3.457	<0.001		
RUE_TP	Species richness (SR)	0.05	0.039	1.29	0.201	0.085	0.038
	FRic	0.024	0.039	0.61	0.544		

## Data Availability

The data supporting this study’s findings are available from the corresponding authors upon reasonable request. Data are contained within the article. These data can be found here: https://1drv.ms/f/s!Am0Eh7ccPZBebnKISoMTtI1WjhA?e=CcfnaK, accessed on 25 December 2023.

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
