# Peer review of "Effects of Aquatic Plant Coverage on Diversity and Resource Use Efficiency of Phytoplankton in Urban Wetlands: A Case Study in Jinan, China"

_biology, 2024, doi:10.3390/biology13010044_

Round 1
Reviewer 1 Report
Comments and Suggestions for Authors
The manuscript describes a study of the mutual influence of aquatic plants and phytoplankton in a wetlands in northern China. The paper provides important conclusions for management activities, namely, which aquatic plants should be increased for a more ecologically sustainable ecosystem of a water body. However, the quality of the description of the results is low. In addition, the research methodology is not detailed enough, especially in the aspect of phytoplankton study. Therefore, it seems to me that after the authors have more carefully rewritten the Materials and Methods, Results and corrected the rest of the manuscript, it can be accepted for publication in Biology.
In the following, I present my questions and comments on the manuscript.
Firstly, very striking is the caption throughout the whole text of the manuscript: Error! Reference source not found. This unfortunately gives the impression that the authors did not look at what they were submitting. In addition, in many parts of the text, the lack of references loses the whole meaning of the text. Therefore, authors should check the citation of references throughout the manuscript.
In my opinion, the concept of phytoplankton RUE should be described in more detail in the introduction, as this concept is explained only at the end of the materials and methods.
L. 47 Instead of "Aquatic plants can purify water quality", it would be better to write "Aquatic plants can purify water" or "can improve water quality"
L. 89 The abbreviation FRic needs to be explained.
Line 123 is redundant.
L.126 A sampling depth of 0.5 m is specified. What were the depths of the water bodies?
L. 131 "The technical terms used are explained for clarity" What does this sentence refer to?
L. 136 It does not seem very reliable to determine phytoplacton species based on a single source.
L. 134 What is the phytoplankton counting frame? It would be good to describe in more detail.
L.189, 190 The formulas need to be numbered. It needs to describe what TN, TP, and q are.
The title of Table 1 would sound better if it were renamed "Number of samples and groups of aquatic vegetation coverage".
Subsection 3.1: the word different is written twice
What is the meaning of the first sentence of subsection 3.1? It seems to me that this subsection could start with the second sentence.
At the beginning of the Results section, a description of what plants were found, how many species, phyla, genera is very lacking. For phyntoplankton also, it is necessary to provide representatives of which phyla were found, how many species or at least genera. In the manuscript we first suddenly meet cyanobacteria already in section 3.2.
L. 241 The phrase "as the dominant functional groups in this period" is repeated.
Table 4 should be repositioned in front of Figures 2 and 3.
Table 5 appears to contain redundant characters.
In the manuscript there is no information on whether aquatic plants are fouled by benthic microalgae and cyanobacteria. I think their contribution should also be taken into account.
Comments on the Quality of English LanguageThe manuscript often contains cumbersome convoluted sentences. There is a lack of clarity in the titles of tables and figures.
Author Response
Response
Dear reviewers:
Thank you for your careful review and constructive suggestions regarding our manuscript. The research methodology section has been carefully rewritten to include more information about the phytoplankton study. In addition, we have carefully reviewed and refined the results section to enhance its clarity and quality. We engaged a professional editing service to thoroughly polish the manuscript to ensure clarity and consistency. Changes in the revised manuscript are highlighted in yellow. The main corrections in the manuscript and the responses to the editor’s comments are as follows.
Answer to reviewer 1
Question1: “Firstly, very striking is the caption throughout the whole text of the manuscript: Error! Reference source not found. This unfortunately gives the impression that the authors did not look at what they were submitting. In addition, in many parts of the text, the lack of references loses the whole meaning of the text. Therefore, authors should check the citation of references throughout the manuscript.
Answer by authors to Q1:
Thank you for pointing out the reference problem in our manuscript. We sincerely apologize for the oversight and understand that it may have given the impression that we were careless. Regarding the "Error! Reference source not found", this was an unintended consequence of our mishandling of the cross-references to the figures and tables during the file formatting process. We have now carefully reviewed the entire manuscript and corrected all areas where this error occurred. In addition, we recognize the importance of references in supporting our research and the overall integrity of the manuscript. We have carefully reviewed the entire text to ensure that all necessary references are accurately cited, and we have added additional references where they were lacking.
Error modification regarding cross-referencing of figures and tables: Please see Page 3, line 120, page 7, line 245, page 7, line 259, page 10, line 276, page 12, line 328, page 15, line 408, page 4-5, lines 176-177, page 5, line 187, page 6, line 224, page 7, line 249, page 7, line 254, page 10, lines 295-297, page 10, line 302, page 11, line 313 and page 15, lines 436-437. Page 10, lines 295-297, Page 10, line 302, Page 11, line 313 and Page 15, lines 436-437.
Following your comments, we have added relevant references. Please see line 22, line 28, line 29, line 30, line 31, lines 34-36, and line 41 in the revised manuscript.
Question2: “In my opinion, the concept of phytoplankton RUE should be described in more detail in the introduction, as this concept is explained only at the end of the materials and methods.”
Answer by authors to Q2: Thank you for your helpful comments on the explanation of the concept of resource utilization efficiency (RUE) in our manuscript. We have revised the introduction to provide a detailed explanation of RUE. I hope this addition could ensure that the concept is clearly understood by the reader before discussing it in the context of our methods and results. Please see page 2, lines 67-77 in the revised manuscript.
Question3: “L 47 Instead of "Aquatic plants can purify water quality", it would be better to write "Aquatic plants can purify water" or "can improve water quality"
Answer by authors to Q3: We agree that the original phrase "Aquatic plants can purify water quality" was not as clear or accurate as it could be. Following your recommendation, we have revised this sentence to "Aquatic plants can purify water.". Please see page 2, line 48 in the revised manuscript.
Question4: “L. 89 The abbreviation FRic needs to be explained.”
Answer by authors to Q4: Following your comments, we provide an explanation for the abbreviation FRic in the revised manuscript. Please see page 2, line 95 in the revised manuscript.
Question5: “Line 123 is redundant.”
Answer by authors to Q5: We apologize for the negligence and we have removed the blank lines.
Question6: “L.126 A sampling depth of 0.5 m is specified. What were the depths of the water bodies?”
Answer by authors to Q6: We apologize that the previous description was very unclear and we have added the average water depths. Please see page 4, lines 131-133 in the revised manuscript.
Question7: “L. 131 "The technical terms used are explained for clarity" What does this sentence refer to?”
Answer by authors to Q7: We sincerely appreciate the editor’s attention to detail and apologize for the oversight in omitting this statement during editing. This sentence has now been deleted from the manuscript, and the paragraph has been thoroughly rewritten to avoid confusion. Please see page 4, lines 130-144 in the revised manuscript.
Question8: “L. 136 It does not seem very reliable to determine phytoplacton species based on a single source.”
Answer by authors to Q8: We sincerely appreciate your constructive suggestions. We previously removed some references on phytoplankton identification due to reference count considerations. Following your suggestions, we have added these missing references in the manuscript to ensure the completeness of the literature support. Please see page 4, lines 143-144 in the revised manuscript.
Question9: “L. 134 What is the phytoplankton counting frame? It would be good to describe in more detail.”
Answer by authors to Q9: Thank you very much for your comments, we have revised the manuscript to describe the counting frame in more detail. Please see page 4, lines 140-143 in the revised manuscript.
Question10: “L.189, 190 The formulas need to be numbered. It needs to describe what TN, TP, and q are.”
Answer by authors to Q10: Thank you for your careful review of our manuscript and your specific suggestions. We have added numbers to all formulas. Regarding the TN, TP, and q variables that you mentioned, we have provided clear definitions for these variables. Please see page 5, lines 195-196 in the revised manuscript.
Question11: “The title of Table 1 would sound better if it were renamed "Number of samples and groups of aquatic vegetation coverage".”
Answer by authors to Q11: Thank you for your suggestion regarding the title of Table 1. In line with your recommendation, we have revised the title to " Number of samples and groups of aquatic vegetation coverage.". Please see page 5, line 179 in the revised manuscript.
Question12: “Subsection 3.1: the word different is written twice”
Answer by authors to Q12: Thank you for your careful reading of our manuscript and for pointing out the error in Subsection 3.1. We have corrected the repeated use of the word "different". Please see page 6, line 222 in the revised manuscript.
Question13: “What is the meaning of the first sentence of subsection 3.1? It seems to me that this subsection could start with the second sentence.”
Answer by authors to Q13: We appreciate your constructive comments on subsection 3.1 and the overall results writing. Following your suggestions, we have removed the first sentence and added one sentence to introduce the description of this result. In addition, we have included a concluding sentence at the end of this section. Please see page 6, lines 223-224, and lines 231-233 in the revised manuscript.
Question14: “At the beginning of the Results section, a description of what plants were found, how many species, phyla, genera is very lacking. For phyntoplankton also, it is necessary to provide representatives of which phyla were found, how many species or at least genera. In the manuscript we first suddenly meet cyanobacteria already in section 3.2.”
Answer by authors to Q14: We appreciate your valuable suggestion. We acknowledge that our study lacked a detailed introduction to the phytoplankton discovered. We have modified the beginning of Section 3.2 to comprehensively enumerate and describe the phytoplankton found in the wetlands. Please see page 7, lines 238-245 in the revised manuscript.
Question15: “L. 241 The phrase "as the dominant functional groups in this period" is repeated.”
Answer by authors to Q15: Thank you for pointing out the repetition in Line 241 regarding the phrase "as the dominant functional groups in this period." We appreciate your attention to detail. I acknowledge that this repetition was an oversight in our writing. We have revised the manuscript and removed the redundant phrase. Please see page 7, line 255 in the revised manuscript.
Question16: “Table 4 should be repositioned in front of Figures 2 and 3.”
Answer by authors to Q16: Thank you for your suggestion, we have adjusted the position of Table 4. Please see page 7, lines 264 in the revised manuscript.
Question17: “Table 5 appears to contain redundant characters.”
Answer by authors to Q17: Thank you for your careful review. After rechecking Table 5, we found and removed the unintentionally included redundant characters. Please see page 8, line 266 in the revised manuscript.
Question18: “In the manuscript there is no information on whether aquatic plants are fouled by benthic microalgae and cyanobacteria. I think their contribution should also be taken into account.”
Answer by authors to Q18: Thank you for your comments regarding the potential impacts of benthic microalgae and cyanobacteria on aquatic plants in our study. We appreciate your insightful comments and recognize the importance of this aspect to understanding aquatic ecosystems. We agree that contamination of aquatic plants by benthic microalgae species and cyanobacteria may be an important factor in wetland ecology. We recognize this as a limitation of our study. Your suggestions highlight an important area for future research. We plan to take this into account in subsequent studies, as it will undoubtedly contribute to a more complete understanding of wetland ecosystems. We have made some additions to the 'conclusion' section of the manuscript to recognize this and suggest it as a potential area for further research. Please see page 16, lines 469-473 in the revised manuscript.
Special thanks to you for your good comments.

Reviewer 2 Report
Comments and Suggestions for Authors
Review for the paper "Effects of aquatic plant coverage on diversity and resource use efficiency of phytoplankton in urban wetlands in north of China" by Hongjingzheng Jiang, Aoran Lu, Jiaxin Li, Mengdi Ma, Ge Meng, Qi Chen and Xuwang Yin submitted to "Biology".
General comment.
Phytoplankton are well-known for their diverse and abundant communities in aquatic ecosystems. However, certain aspects of their ecology have received less attention in comparison to classical studies on community structure and seasonal cycles. Specifically, the effects of macrophytes on phytoplankton have been relatively understudied. Phytoplankton, as primary producers, regulate energy fluxes in freshwater environments. Consequently, assessing the impact of macrophytes on their diversity poses a significant challenge. This paper explores phytoplankton assemblages in urban wetlands in North China. The authors demonstrate a direct relationship between macrophyte coverage levels, phytoplankton diversity and composition, and functional groups. The research has the potential to interest specialists in aquatic ecology and management. Overall, the paper is well-written, but there are numerous editorial and formatting errors that must be corrected.
Specific comments.
L88. Consider replacing "This study studied" with "This study examined" .
L97-117. It is suggested to split the section into 2 paragraphs (L97-107 and L107-117).
L108. Provide depth ranges in the wetlands sampled and total number of phytoplankton samples.
L113-114, 169, 180, 239, 262, Section 3.4., 311, 393, 422. Check the reference.
L137. Indicate the units for biomass (dry or wet).
L161. Provide a reference for determining plant coverage
L196. Consider replacing 'Origin 2023' with 'the software Origin 2023'?
L230. Delete 'results'.
Discussion. The authors should include a section that considers additional factors that may be responsible for the interactions between macrophytes and phytoplankton. Specifically, the impact of water temperature should be examined.
Comments on the Quality of English LanguageSome revisions are required.
Author Response
Response
Dear reviewers:
Thank you very much for your positive comments on our manuscript, as well as the editorial and formatting issues you pointed out. We have carefully reviewed the entire manuscript and corrected the various issues that arose. We engaged a professional editing service to thoroughly polish the manuscript to ensure clarity and consistency. Changes in the revised manuscript are highlighted in green. The major revisions in the manuscript and the responses to the editor’s comments are as follows.
Answer to reviewer 2
Question1: “L88. Consider replacing "This study studied" with "This study examined”
Answer by authors to Q1: Thank you for your attentive reading and the suggestion to refine the wording in our manuscript. In line 88, we have replaced "This study studied" with "This study examined" as you recommended. Please see page 2, line 94 in the revised manuscript.
Question2: “L97-117. It is suggested to split the section into 2 paragraphs (L97-107 and L107-117).”
Answer by authors to Q2: Thank you for your constructive suggestion. Following your advice, we have divided this portion of the manuscript into two distinct paragraphs. The first paragraph now covers lines 103-113, and the second paragraph comprising lines 114-123. Please see page 3, lines 103-123 in revised manuscript.
Question3: “L108. Provide depth ranges in the wetlands sampled and total number of phytoplankton samples."
Answer by authors to Q3: We sincerely appreciate your detailed review of our manuscript and the invaluable suggestions you have provided. Significant revisions have been made to the phytoplankton collection section of the manuscript to supplement many details that were previously omitted. Please see page 4, lines 130-136 in the revised manuscript.
Question4: “L113-114, 169, 180, 239, 262, Section 3.4., 311, 393, 422. Check the reference.”
Answer by authors to Q4: Thank you for pointing out the reference problem in our manuscript. Regarding the "Error! Reference source not found", this was an unintended consequence of our mishandling of the cross-references to the figures and tables during the file formatting process. We have now carefully reviewed the entire manuscript and corrected all areas where this error occurred. Error modification regarding cross-referencing of figures and tables: Please see Page 3, line 120, page 7, line 245, page 7, line 259, page 10, line 276, page 12, line 328, page 15, line 408, page 4-5, lines 176-177, page 5, line 187, page 6, line 224, page 7, line 249, page 7, line 254, page 10, lines 295-297, page 10, line 302, page 11, line 313 and page 15, lines 436-437. Page 10, lines 295-297, Page 10, line 302, Page 11, line 313 and Page 15, lines 436-437.
Question5: “L137. Indicate the units for biomass (dry or wet).”
Answer by authors to Q5: Thank you for your careful review of our manuscript. Regarding the biomass unit issue you pointed out, we used the wet weight of phytoplankton as the unit of measurement for biomass. Please see page 4, line 144 in the revised manuscript.
Question6: “L161. Provide a reference for determining plant coverage”
Answer by authors to Q6: Thank you for your guidance, we have added relevant reference sources.
Cui xinhong; Chen jiakuan; Li wei SURVEY METHODS ON AQUATIC MACROPHYTE VEGETATION IN LAKES IN THE MIDDLE AND LOWER REACHES OF CHANGJIANG RIVER. Journal of Wuhan Botanical Research 1999, 357–361(崔心红,陈家宽,李伟.长江中下游湖泊水生植被调查方法[J].武汉植物学研究,1999(04):357-361.).
Please see page 5, line 169 in the revised manuscript.
Question7: “L196. Consider replacing 'Origin 2023' with 'the software Origin 2023'?”
Answer by authors to Q7: Thank you for your suggestion regarding the description of "Origin 2023" in our manuscript. We have made this change in our revised manuscript. Please see page 6, line 202 in the revised manuscript.
Question8: “L230. Delete 'results'.”
Answer by authors to Q8: Thanks to your suggestion, we have removed the word "results" from line 230 of the original manuscript. Please see page 7, line 247 in the revised manuscript.
Question9: “Discussion. The authors should include a section that considers additional factors that may be responsible for the interactions between macrophytes and phytoplankton. Specifically, the impact of water temperature should be examined.”
Answer by authors to Q9: Thank you for your insightful comments on our study in analyzing interactions between macrobenthos and phytoplankton. We recognize that water temperature plays an important role in influencing phytoplankton dynamics and macrobenthic growth. In our preliminary draft, water temperature was used as one of the observed variables that make up the latent variables (environmental factors). However, the results of PLS-SEM modeling analysis showed that an increase in aquatic plant cover significantly increased the concentration of total nitrogen in the water column, which is contrary to our water quality monitoring results. During the data validation process, we identified a critical issue related to temperature data. The water temperature measurements we had were not collected from the same observation point at the same time each day. Instead, they represented water temperatures at completely different times of the week, which led to significant inconsistencies. Given the importance of water temperature and its significant impact, we decided to exclude this variable from the analysis to maintain the integrity and reliability of the results. We recognize this as a limitation of our current study and thank you for bringing it to our attention. In our next steps, we plan to adopt a more rigorous approach to data collection to ensure temporal and locational consistency of temperature data.
Special thanks to you for your good comments.

Reviewer 3 Report
Comments and Suggestions for Authors
Wetlands in urban areas have multiple implications for the population and ecological stability of the regions. Maintaining water quality in urban wetlands is a challenge for the population of many regions of the world. The authors of the MS carried out an important study on the influence of biotic factors on water quality in wetlands of Shandong Province, China; moreover, the conclusions drawn can be applied to freshwater open reservoirs around the world.
The manuscript can be accepted after the authors have made a number of corrections.
Section 2.2: Please indicate the number of samples.
Definition of the term “resource use efficiency” (RUE) should be made on line 45 when first mentioned, and not on line 69.
Lines 144-146: “Categorization of phytoplankton was based on genus affiliation among the 39 FGs of the classification system [30].” However in [30] only 7 functional groups are indicated.. In the Line 237 “phytoplankton can be classified into 24 functional groups” but there are 28 groups shown in Table 4.
Lines 169, 180, 217, 234, 239, 262 -delete (Error! Reference source not found.)
Line 180: add abbreviation FRic.
Lines 234, 239, 262, 280-283, 286, 311: сhange (Error! Reference source not found.) to the relevant links.
Line 181: change “in reference to pertinent literature [34]” to “as described [34]”
Table 2: change:
|
Biovolume |
<100 μm3, 100–1,000μm3, 1,000–10,000μm3,>10,000 lm3 |
to
|
Biovolume (μm3/m3) |
<100, 100–1,000, 1,000–10,000, >10,000 |
Line 182: Refs[ 66] and [67] are located immediately after [34].
Line 200: move (Origin Lab, California, United States) to Line 196
Lines 240-241: “as the dominant functional groups in this period” – delete duplication
Section 3.2: use the previously entered abbreviation FG
Table 4: add spp. to Ochromonas and Chrysochromulina
Lines 137, 229, 256: change “FG functional group” to “functional groups”
Line 265 tenue – change font to Italic
Line 376: Ref [79] is absent in a Reference list
Line 439: Ref [90] is absent in a Reference list
Line 446: Ref [61] preceded Ref [63]
Refs [51, 52] are absent in the text
Ref [46] - not available
Ref [48] – incomplete Ref.
Author Response
Response
Dear reviewers:
Thank you for your careful review and constructive suggestions regarding our manuscript. We appreciate your pointing out the problems with reference citations and formatting and have corrected them following your comments. We engaged a professional editing service to thoroughly polish the manuscript to ensure clarity and consistency. Changes in the revised manuscript are highlighted in blue-green. The main corrections in the manuscript and the responses to the editor’s comments are as follows.
Answer to reviewer 3
Question1: “Section 2.2: Please indicate the number of samples”.
Answer by authors to Q1: We appreciate your attention to detail and we have rewritten the phytoplankton collection section to add the missing detail section. Please see page 4, lines 130-136 in the revised manuscript.
Question2: “Definition of the term “resource use efficiency” (RUE) should be made on line 45 when first mentioned, and not on line 69.”
Answer by authors to Q2: Thank you for your careful review of our manuscript. Based on your suggestions, we have revised our manuscript to change the definition of "resource use efficiency (RUE)" from line 69 to lines 45-46. Please see page 2, lines 45-46 in the revised manuscript.
Question3: “Lines 144-146: “Categorization of phytoplankton was based on genus affiliation among the 39 FGs of the classification system [30]. However in [30] only 7 functional groups are indicated. In the Line 237 “phytoplankton can be classified into 24 functional groups” but there are 28 groups shown in Table 4."
Answer by authors to Q3: We deeply appreciate your thorough review of our manuscript. Regarding your comment on lines 144-146, this was due to an oversight in our citation of references. This has been corrected in the revised manuscript and detailed references supplementing the 39 functional group classifications have been added. As for the inconsistency between line 237 and Table 4 you pointed out, we acknowledge this error and thank you for your attention to detail. The correct statement should be "phytoplankton can be classified into 28 functional groups. We have revised line 237 accordingly and ensured consistency with Table 4. Please see page 4, lines 153-154, and page 7, line 252 in the revised manuscript.
Question4: “Lines 169, 180, 217, 234, 239, 262 -delete (Error! Reference source not found.)”
Answer by authors to Q4: Thank you for pointing out the reference problem in our manuscript. Regarding the "Error! Reference source not found", this was an unintended consequence of our mishandling of the cross-references to the figures and tables during the file formatting process. We have now carefully reviewed the entire manuscript and corrected all areas where this error occurred. Error modification regarding cross-referencing of figures and tables: Please see Page 3, line 120, page 7, line 245, page 7, line 259, page 10, line 276, page 12, line 328, page 15, line 408, page 4-5, lines 176-177, page 5, line 187, page 6, line 224, page 7, line 249, page 7, line 254, page 10, lines 295-297, page 10, line 302, page 11, line 313 and page 15, lines 436-437. Page 10, lines 295-297, Page 10, line 302, Page 11, line 313 and Page 15, lines 436-437.
Question5: “Line 180: add abbreviation FRic.”
Answer by authors to Q5: Thank you for your suggestion to add the abbreviation "FRic" at line 180 of our manuscript. We have now incorporated "FRic" at the specified location in the manuscript. Please see page 5, line 187 in the revised manuscript.
Question6: “Lines 234, 239, 262, 280-283, 286, 311: сhange (Error! Reference source not found.) to the relevant links.”
Answer by authors to Q6: Thank you for pointing out the reference problem in our manuscript. Regarding the "Error! Reference source not found", this was an unintended consequence of our mishandling of the cross-references to the figures and tables during the file formatting process. We have now carefully reviewed the entire manuscript and corrected all areas where this error occurred. Error modification regarding cross-referencing of figures and tables: Please see Page 3, line 120, page 7, line 245, page 7, line 259, page 10, line 276, page 12, line 328, page 15, line 408, page 4-5, lines 176-177, page 5, line 187, page 6, line 224, page 7, line 249, page 7, line 254, page 10, lines 295-297, page 10, line 302, page 11, line 313 and page 15, lines 436-437. Page 10, lines 295-297, Page 10, line 302, Page 11, line 313 and Page 15, lines 436-437.
Question7: “Line 181: change “in reference to pertinent literature [34]” to “as described [34]””
Answer by authors to Q7: We appreciate your attention to detail and we have changed “about pertinent literature [34]” to “as described [34]”. Please see page 5, line 188 in the revised manuscript.
Question8: “Table 2: change:”
Answer by authors to Q8: Thank you for your detailed feedback on the formatting of the tables in our manuscript. The specific parts of the table that you have pointed out have been modified based on your suggestions. Please see page 5, line 189 in the revised manuscript.
Question9: “Line 182: Refs[ 66] and [67] are located immediately after [34].”
Answer by authors to Q9: We deeply appreciate you identifying the improperly ordered references in our manuscript. This issue occurred because we failed to update the citation order in time during the final editing process. We have now thoroughly reviewed the entire paper to ensure all references are ordered correctly. Please see page 5, line 189 in the revised manuscript.
Question10: “Line 200: move (Origin Lab, California, United States) to Line 196”
Answer by authors to Q10: We appreciate your attention to detail in our manuscript. As suggested, we have moved the phrase "(Origin Lab, California, United States)" to the position where the software name first appears. Please see page 6, lines 202-203 in the revised manuscript.
Question11: “Lines 240-241: “as the dominant functional groups in this period” – delete duplication”
Answer by authors to Q11: We apologize for the oversight and thank you for pointing out the repetitive phrase in lines 240-241 of our manuscript. We have removed the redundant phrase from this section in the revised manuscript. Please see page 7, lines 255-256 in the revised manuscript.
Question12: “Section 3.2: use the previously entered abbreviation FG”
Answer by authors to Q12: Thank you for your attention to detail on our manuscript. We have not only updated the content in Section 3.2 but also modified the title to better reflect the revised content. Please see page 7, line 237 in the revised manuscript.
Question13: “Table 4: add spp. to Ochromonas and Chrysochromulina”
Answer by authors to Q13: We appreciate your careful review of the tables in our manuscript. As you have correctly indicated, we have added "spp." after the genus names "Ochromonas" and "Chrysochromulina" in Table 4. Please see page 7, lines 264-265 in the revised manuscript.
Question14: “Lines 137, 229, 256: change “FG functional group” to “functional groups”
Answer by authors to Q14: Thank you for your careful review. we have changed “FG functional group” to “functional groups” in lines 137, 229, and 256. Please see page 4, line 146, page 7, line 237, and page 7, line 264 in the revised manuscript.
Question15: “Line 265 tenue – change font to Italic”
Answer by authors to Q15: Thank you for your attention to detail in our manuscript. Following your suggestion, we have changed the font of "tenue" to Italic on page 10, line 280.
Question16: “Line 376: Ref [79] is absent in a Reference list”
Answer by authors to Q16: We appreciate you pointing out many problems with the reference list in our manuscript. We sincerely apologize for this oversight. These are because our citation software fails to timely update the reference list. We thoroughly cross-checked all 76 references cited in the manuscript and ensured they were accurately reflected in the reference list. In the process, we also reviewed and corrected any errors or inconsistencies in the citation of the references in the manuscript.
Question17: “Line 439: Ref [90] is absent in a Reference list”
Answer by authors to Q17: We appreciate you pointing out many problems with the reference list in our manuscript. We sincerely apologize for this oversight. These are because our citation software fails to timely update the reference list. We thoroughly cross-checked all 76 references cited in the manuscript and ensured they were accurately reflected in the reference list. In the process, we also reviewed and corrected any errors or inconsistencies in the citation of the references in the manuscript.
Question18: “Line 446: Ref [61] preceded Ref [63]”
Answer by authors to Q18: We appreciate you pointing out many problems with the reference list in our manuscript. We sincerely apologize for this oversight. These are because our citation software fails to timely update the reference list. We thoroughly cross-checked all 76 references cited in the manuscript and ensured they were accurately reflected in the reference list. In the process, we also reviewed and corrected any errors or inconsistencies in the citation of the references in the manuscript.
Question19: “Refs [51, 52] are absent in the text”
Answer by authors to Q19: We appreciate you pointing out many problems with the reference list in our manuscript. We sincerely apologize for this oversight. These are because our citation software fails to timely update the reference list. We thoroughly cross-checked all 76 references cited in the manuscript and ensured they were accurately reflected in the reference list. In the process, we also reviewed and corrected any errors or inconsistencies in the citation of the references in the manuscript.
Question20: “Ref [46] - not available”
Answer by authors to Q20: We appreciate you pointing out many problems with the reference list in our manuscript. We sincerely apologize for this oversight. These are because our citation software fails to timely update the reference list. We thoroughly cross-checked all 76 references cited in the manuscript and ensured they were accurately reflected in the reference list. In the process, we also reviewed and corrected any errors or inconsistencies in the citation of the references in the manuscript.
Question21: “Ref [48] – incomplete Ref.”
Answer by authors to Q21: We appreciate you pointing out many problems with the reference list in our manuscript. We sincerely apologize for this oversight. These are because our citation software fails to timely update the reference list. We thoroughly cross-checked all 76 references cited in the manuscript and ensured they were accurately reflected in the reference list. In the process, we also reviewed and corrected any errors or inconsistencies in the citation of the references in the manuscript.
Special thanks to you for your good comments.

Round 2
Reviewer 1 Report
Comments and Suggestions for Authors
The authors have provided full answers to all my questions on the manuscript Effects of aquatic plant coverage on diversity and resource use efficiency of phytoplankton in urban wetlands in north of China. All my comments have been addressed. As a result of the corrections made by the authors, the manuscript has been greatly improved and is now deserving to be published in Biology.
The only comment: capital letters are used in the reference list under numbers 2, 35, 41.
Author Response
Response
Dear reviewers:
Thank you for your careful review and constructive suggestions regarding our manuscript. The research methodology section has been carefully rewritten to include more information about the phytoplankton study. In addition, we have carefully reviewed and refined the results section to enhance its clarity and quality. We engaged a professional editing service to thoroughly polish the manuscript to ensure clarity and consistency. Changes in the revised manuscript are highlighted in yellow. The main corrections in the manuscript and the responses to the editor’s comments are as follows.
Answer to reviewer 1
Question1: “The only comment: capital letters are used in the reference list under numbers 2, 35, 41”.
Answer by authors to Q1:
Thank you for pointing out the reference problem in our manuscript. We have changed the citation issue for references. Please see page 17, lines 526-527, page 19, line 606, and lines 619-620 in the revised manuscript.